# Experiences of Migrant People Living with HIV in a Multidisciplinary HIV Care Setting with Rapid B/F/TAF Initiation and Cost-Covered Treatment: The ‘ASAP’ Study

**DOI:** 10.3390/jpm12091497

**Published:** 2022-09-13

**Authors:** Anish K. Arora, Kim Engler, David Lessard, Nadine Kronfli, Adriana Rodriguez-Cruz, Edmundo Huerta, Benoit Lemire, Jean-Pierre Routy, René Wittmer, Joseph Cox, Alexandra de Pokomandy, Lina Del Balso, Marina Klein, Giada Sebastiani, Isabelle Vedel, Amélie Quesnel-Vallée, Bertrand Lebouché

**Affiliations:** 1Department of Family Medicine, Faculty of Medicine & Health Sciences, McGill University, Montréal, QC H3S 1Z1, Canada; 2Centre for Outcomes Research & Evaluation, Research Institute of the McGill University Health Centre, Montréal, QC H4A 3S5, Canada; 3Infectious Diseases and Immunity in Global Health Program, Research Institute of the McGill University Health Centre, Montréal, QC H4A 3S5, Canada; 4Canadian Institutes of Health Research Strategy for Patient-Oriented Research (CIHR/SPOR) Mentorship Chair in Innovative Clinical Trials in HIV Care, Montréal, QC H4A 3S5, Canada; 5Department of Medicine, Chronic Viral Illness Service, Division of Infectious Diseases, McGill University Health Centre, Montréal, QC H4A 3J1, Canada; 6Pharmacy Department, McGill University Health Centre, Montréal, QC H4A 3J1, Canada; 7Department of Family Medicine and Emergency Medicine, Université de Montréal, Montréal, QC H3C 3J7, Canada; 8Department of Epidemiology, Biostatistics and Occupational Health, Faculty of Medicine & Health Sciences, McGill University, Montréal, QC H3A 1A2, Canada; 9Department of Sociology, Faculty of Arts, McGill University, Montréal, QC H3A 0G5, Canada

**Keywords:** HIV, migrants, B/F/TAF, antiretroviral, rapid ART initiation, multidisciplinary, cost-covered treatment, HIV care cascade, patient experiences

## Abstract

This study aimed to explore the experiences of migrant people living with HIV (MLWH) enrolled in a Montreal-based multidisciplinary HIV care clinic with rapid antiretroviral treatment (ART) initiation and cost-covered ART. Between February 2020 and March 2022, 32 interviews were conducted with 16 MLWH at three time-points (16 after 1 week of ART initiation, 8 after 24 weeks, 8 after 48 weeks). Interviews were analyzed via the Framework Method. Thirty categories were identified, capturing experiences across the HIV care cascade. At diagnosis, most MLWH described “initially experiencing distress”. At linkage, almost all MLWH discussed “navigating the health system with difficulty”. At treatment initiation, almost all MLWH expressed “being satisfied with treatment”, particularly due to a lack of side effects. Regarding care retention, all MLWH noted “facing psychosocial or health-related challenges beyond HIV”. Regarding ART adherence, most MLWH expressed “being satisfied with treatment” with emphasis on their taking control of HIV. At viral suppression, MLWH mentioned “finding more peace of mind since becoming undetectable”. Regarding their perceived health-related quality of life, most MLWH indicated “being helped by a supportive social network”. Efficient, humanizing, and holistic approaches to care in a multidisciplinary setting, coupled with rapid and free ART initiation, seemed to help alleviate patients’ concerns, address their bio-psycho-social challenges, encourage their initial and sustained engagement with HIV care and treatment, and ultimately contribute to positive experiences.

## 1. Introduction

Migrant people living with HIV (MLWH) are a diverse, growing, and at times a vulnerable and/or marginalized population in Canada and other countries affiliated with the Organization for Economic Co-operation and Development [1,2,3]. Across these countries, MLWH often experience delayed entry into HIV care, late antiretroviral treatment (ART) initiation, higher rates of care drop-out, poorer adherence to ART, and variable rates of viral suppression, when compared to native-born populations living with HIV [1,2,3,4,5,6,7,8,9]. These issues are, in part, due to the numerous multilevel barriers that MLWH experience across their HIV care trajectory [10,11,12,13,14,15,16,17,18,19,20,21,22].

To address barriers faced by MLWH, who have diverse ethnic, geographic, and cultural origins, and who may often experience precarious legal and health coverage statuses, targeted interventions may be necessary [3]. In fact, HIV scholars who call for an equity-focused approach to ending the HIV epidemic point out that when efforts are targeted to specific populations with the heaviest burden of HIV, there is a greater potential for improved population health and lower HIV transmission rates [23]. Targeted interventions which specifically respond to population needs and associated social determinants of health (e.g., housing, poverty, and other barriers to care based on structural racism) may also be economically favorable [23]. Multidisciplinary care models with sufficient funding for social workers or staff with similar training and expertise (e.g., case managers) may facilitate the development and implementation of targeted interventions. These professionals can assist MLWH in addressing specific individual-level challenges (e.g., immigration, obtaining health coverage, finding a job, mental health issues) which require resolution to allow for long-term engagement with HIV care and treatment [22]. 

Rapid ART initiation has been endorsed as another strategy to efficiently engage populations in HIV care and treatment, particularly those who are vulnerable or marginalized [24,25,26,27,28,29,30,31]. Rapid ART initiation is defined as linkage to care and start of ART as soon as possible after a new HIV diagnosis [30]. Major advancements in ART over the last two decades have given rise to biological and clinical benefits for people living with HIV, alongside public health benefits for populations and public health systems in general [24,25,26,27,28,29,30]. Moreover, rapid ART initiation has been shown to reduce loss-to-follow-up between HIV testing and treatment initiation, improve retention in care, and reduce time to viral suppression, without compromising safety [25,26,28,29,30,31]. However, to ensure that potentially vulnerable and marginalized groups such as MLWH can experience sustained benefit from rapid ART initiation, additional resources and support are recommended [27]. Using a multidisciplinary model of HIV care with rapid ART initiation may assist in addressing underlying challenges that prevent patients from starting and remaining engaged in care and treatment, while also effectively responding to emerging priorities for HIV service delivery [32]. Additionally, having pharmacists embedded in the multidisciplinary model of HIV care may facilitate rapid ART initiation through easier and more efficient access to treatment and thorough consideration of drug–drug interactions. Moreover, if ART is provided free-of-charge, this may further reduce challenges faced by MLWH with treatment initiation (e.g., lack of health coverage).

Importantly, although the clinical and public health importance of rapid ART initiation and multidisciplinary HIV care have been well described in the literature, the experiences of MLWH around such care models have rarely been explored, especially in the context of cost-covered treatment. Exploring patient experience is important to understand how services are received by patients and how they could be improved to better meet their needs [33,34]. This information may also help orient priorities within clinics or across health systems [33,34]. Thus, the purpose of this study is to explore and document the experiences of MLWH who are enrolled in a care model comprising multidisciplinary HIV care, rapid ART initiation, and cost-covered treatment.

## 2. Materials and Methods

### 2.1. Research Question

The following question guided this research endeavor: what are the care experiences of MLWH enrolled in multidisciplinary HIV care with free and rapid initiation of ART?

### 2.2. Design

In January 2020, we initiated a 96-week prospective cohort study (the ‘ASAP’ study) with a convergent mixed-method design at the Chronic Viral Illness Service of the McGill University Health Centre (CVIS/MUHC), a quaternary hospital-based clinic serving the largest proportion of MLWH in Montreal, Canada. As of June 2022, 40 patients were enrolled, of whom 30 were migrants. All participating patients were provided care by a multidisciplinary team composed of on-site physicians, nurses, social workers, pharmacists, and a psychiatrist.

All patients received bictegravir/emtricitabine/tenofovir alafenamide (B/F/TAF) as soon as possible (ideally within 7 days) after being linked to care. B/F/TAF was provided free-of-charge for the duration of the study. It is a once-daily single-tablet regimen for the treatment of HIV-1 infection in adults [35,36]. It has a high genetic barrier to the development of resistance, is generally well tolerated, requires no prior HLA-B*5701 testing, fulfils the antiretroviral regimen requirement for patients with hepatitis B virus co-infection, and can be used in renally impaired patients with creatinine clearance ≥ 30 mL/min [35,36]. Moreover, B/F/TAF has few potential drug–drug interactions, a small pill size, no food intake requirements, and no baseline viral load or CD4 cell count restrictions, thus making it suitable for rapid ART initiation [36]. Health Canada approved B/F/TAF as a complete regimen for the treatment of HIV-1 infection in July 2018 [36]. 

Details of the ‘ASAP’ study’s design are reported elsewhere [37]. This manuscript presents an analysis of the study’s qualitative data collected from MLWH up to March 2022.

### 2.3. Data Collection

Semi-structured individual interviews were conducted with MLWH face-to-face, by telephone, or video-conferencing at three time-points: after 1 week of treatment initiation, after 24 weeks, and after 48 weeks. The first interview solicits information on the experience of beginning HIV care and treatment, focusing specifically on participants’ satisfaction, worries, expected benefits, and suggestions for improving HIV services. The second interview initially asks patients to recount their experience of being linked to HIV care and treatment. Patients are then asked to describe how their “general situation” has evolved since beginning HIV care and treatment (with prompts around changes in quality of life, sociodemographics, and access to social services and healthcare). Then, the interview probes the impact of services and staff at the clinic on the participants’ “situation” (i.e., their health, wellbeing, and life in general) and treatment taking. It concludes with a question about their thoughts on their care and treatment (with prompts on the negative and positive aspects, things that could be improved, and related impacts of the immigration process). The third interview asks about responsibility to manage care and treatment, comfort with care providers, impacts on health and lifestyle since initiating care and treatment, and suggestions for improving services at the clinic. See the Appendix A for the full interview guides.

### 2.4. Data Analysis

In the ASAP study, interviews with MLWH were conducted in English, French, and Spanish. Data pertaining only to the interviews conducted in English and French were analyzed for this article as these were the only interviews available for analysis at this time, and because most of our team is fluent in both of these languages. Interviews were transcribed verbatim by a professional transcriber who is fluent in English and French. Transcribed interviews were imported into QSR’s NVIVO 12 and analyzed via the Framework Method [38,39,40,41]. This qualitative method, in use since the 1980s, was originally developed by Richie and Spencer as a pragmatic approach for large-scale social policy research [38,39,40,41]. Over the last four decades, it has been widely taken-up in medical and health research [38]. Gale et al. highlight the appropriateness of this method when engaging in applied health research with large qualitative datasets and multidisciplinary teams that incorporate patients, clinicians, and scientists [38], which is precisely the context in which this study was conducted. The Framework Method provides a highly structured approach to data analysis, akin to content and thematic analysis; however, its defining feature is the presentation of the results through data displays or matrices [38,39,40,41].

The approach by Gale et al. consists of seven stages: (1) transcription, where qualitative data are transcribed; (2) data familiarization, where researchers review the dataset to understand its content and structure and begin interpreting the data and identifying possible codes and patterns; (3) coding, where researchers read transcripts line-by-line and apply a paraphrase or label (a ‘code’) to each line or substantive block of text; (4) developing a working analytical framework, where, after coding the first few transcripts, researchers meet to iteratively compare the codes and discuss possible categories for grouping the codes; (5) applying the analytical framework, where the agreed-upon analytical framework is applied to all the transcripts; (6) charting data into the framework matrix, where summarized data/quotes are added, by category, to a data display or matrix, which is then viewed, revised, and validated by the multidisciplinary team members; and lastly, (7) interpreting the data, where the characteristics of and differences between the data are identified, and, if the data are rich enough, the findings can go beyond description to an explanation of the phenomena [38].

An inductive–deductive approach to analysis was taken in this study. Codes were inductively generated to thoroughly capture the participants’ experiences and were iteratively revised and grouped to generate categories. Categories were then deductively grouped, based on the seven steps of the HIV care cascade (i.e., HIV diagnosis, linkage to care, treatment initiation, retention in care, adherence to treatment, viral suppression, and health-related quality of life) [42,43]. Grouping by HIV care cascade steps occurred based on the way MLWH described their experiences, feelings, or overall thoughts. For example, when participants spoke about their experience of being linked to care, all codes generated in that block of text were ascribed to the linkage step of the cascade. Caution was taken when participants had not experienced a particular HIV care cascade step (e.g., retention in care), but still described their initial feelings and thoughts about that step in the earlier interviews. The first author performed all coding. The codes, categories, and the framework matrix were iteratively reviewed, revised, and validated through research and stakeholder committee meetings held between May 2022 and June 2022. Specifically, seven meetings were held with the research team responsible for the analysis (AKA, KE, BL, and twice with DL); one was held with the first author’s thesis advisory committee (BL, AQV, NK, and IV), two were held one-on-one with two different patient-partners (their names are kept anonymous to protect their identities); one was held with the study coordinator (ED); and an additional meeting was held with a research nurse (LDB). Note that only categories with three or more contributing participants (i.e., a minimum saturation of 19%) are presented in this manuscript to further ensure trustworthiness. 

### 2.5. Patient and Stakeholder Engagement

A patient and stakeholder engagement approach was taken, whereby key stakeholders (i.e., patients, clinicians, and community organization leaders) were involved throughout the research process via advisory committee meetings [44,45,46]. Specifically, three MLWH (a Latin American asylum seeker, a European international student, and an African asylum seeker) receiving care at the study site (CVIS/MUHC), three community representatives, and five CVIS/MUHC healthcare professionals (i.e., two social workers and 3 nurses) provided feedback on the study design, validated the ‘ASAP’ study protocol, and reviewed the interview schedules for acceptability, clarity, and quality [37]. Also, as noted above, two patient-partners, a research nurse, and a research coordinator were involved in data analysis through one-on-one engagement to enable appropriate interpretation of the data for this study.

### 2.6. Ethics

This study was conducted in accordance with applicable Health Canada regulations, International Conference on Harmonisation guidelines on current Good Clinical Practice, and the Declaration of Helsinki. It was approved by the Research Ethics Board of the Research Institute of the McGill University Health Centre (reference #: MP-37-2020-4911). Informed consent was obtained from all study participants.

## 3. Results

A total of 32 semi-structured interviews were conducted with 16 MLWH at three time-points (16 after 1 week of treatment initiation, 8 after 24 weeks, and 8 after 48 weeks) between February 2020 and March 2022. The average duration of the interviews based on the available timestamps (*n* = 26/32; timestamps were unavailable for interviews where participants did not want to be recorded) were: 20 min (range: 15–43 min); 41 min (range: 23–72 min); and 28 min (range: 13–42 min), respectively. 

### 3.1. Participant Demographics at Enrollment 

The participants’ sociodemographic characteristics at enrollment are displayed in Table 1. The average age of the participants was 36 years old (range: 24–55). Most (11/16; 69%) were males who identified as gay or bisexual. Participants were born in Africa (*n* = 6), Asia (*n* = 4), Europe (*n* = 1), Latin America (*n* = 3), and the Caribbean (*n* = 2). Participants had varied immigration statuses in Canada: asylum seeker (7/16; 44%); international student (3/16; 19%); international worker (1/16; 6%); visitor (3/16; 19%), naturalized citizen (1/16; 6%) and undocumented (1/16; 6%). Seven participants (44%) had no or low health coverage, whereas nine (56%) had sufficient coverage (i.e., HIV treatment and care was covered by their insurance). Most had prior university-level education (10/16; 63%), while the remaining participants had either a college diploma (4/16; 25%) or secondary education/professional degree (2/16; 13%). Most were unemployed (10/16; 63%); all others had paid employment (6/16; 38%).

### 3.2. Categories

Through the framework analysis, a total of 30 categories were identified which capture the experiences of MLWH across the HIV care cascade steps while enrolled in multidisciplinary HIV care, with the rapid ART initiation, and cost-covered ART. The categories are presented in quotations and the associated sub-categories are presented in italics. Figure 1 provides a summary of the main categories. A data framework matrix, presented in Table 2, provides illustrative interview excerpts. Table 2 also presents the number of participants who contributed to each category (i.e., out of the 16 participants, how many spoke about each category and sub-category) and the number of interviews that contributed to each category (i.e., out of the 32 interviews, how many had content for each category and sub-category). This information is provided to demonstrate data saturation, both by number of participants and longitudinally (since only half of the participants were able to complete the interviews in weeks 24 and 48). Saturation level by the number of participants is also provided in the text.

#### 3.2.1. HIV Diagnosis

Four categories were ascribed to the HIV diagnosis step of the HIV care cascade. Upon learning of their positive HIV status, most participants (10/16) discussed “initially experiencing distress”, or more specifically, feeling: *worried and/or scared* (7/16), *shocked* (6/16), *confused* (6/16), or *like they lost control of their life* (4/16). Beyond initial distress, a little over a third of the participants (6/16) were “questioning the impact of HIV diagnosis on immigration”, specifically wondering about the effect that their new HIV status would have on their immigration applications. Four participants also discussed how they were “fearing stigmatization”. In this regard, the participants expressed concern about how others would treat them once they found out about their HIV status (e.g., backbiting in the community, being discriminated against by healthcare professionals, being ostracized by family, losing one’s job). Three participants also underscored their “uncertainty about HIV testing requirements for migrants”. These participants explained that, due to variations in policies, certain migrant populations (e.g., temporary visitors or international students from certain regions) were exempt from HIV testing via the Immigration Medical Exam. As a result, testing for HIV was either left to the patients’ judgement or occurred during specialty care for another health issue.

#### 3.2.2. Linkage to HIV Care

Six categories were ascribed to the linkage to care step of the HIV care cascade. While being linked to HIV care, almost all participants (15/16) mentioned that they were “navigating the health system with difficulty”. Almost all participants (15/16) discussed challenges with navigation *across clinics and organizations* (e.g., from the service that diagnosed them to their HIV care center, the CVIS/MUHC). Nine participants discussed challenges specifically *within the CVIS/MUHC* clinic. Difficulties with navigation were attributed to their unfamiliarity with the Québec culture, language, and healthcare system, as well as to confusion around the specific roles and responsibilities of the numerous healthcare professionals they were encountering at care onset. 

An element that most participants (14/16) described as being indispensable to their early engagement with care and treatment was experiencing “humanizing clinical encounters”. This category encompassed five sub-categories: *feeling supported and cared for* (12/16), *feeling kindness from healthcare professionals* (10/16), *feeling safe and comfortable* (7/16), *feeling heard and accepted* (5/16), and *feeling respected* (3/16). Participants also expressed “being reassured about living with HIV” (12/16), which was deemed helpful in calming initial negative emotions and fears. Additionally, most participants (13/16) discussed “receiving personalized health information”, which consisted of adapted answers to their HIV-related health concerns, reference to resources to better manage their health and wellbeing, and advice based on the individual and their lifestyle. Ten participants emphasized their experience of “quickly accessing care”, which was often expressed with appreciation. The final category associated with linkage to HIV care was “facing psycho-social challenges beyond HIV” (12/16). Participants expressed that, beyond their HIV, they dealt with tremendous stressors, including *mental health* (11/16), *immigration* (8/16), *securing finances and/or health insurance* (6/16), and *learning Quebec’s official language* (4/16). 

#### 3.2.3. Treatment Initiation

Four categories were ascribed to the ART initiation step of the HIV care cascade. Almost all participants (15/16) expressed “being satisfied with treatment” within the first week of initiation. Their satisfaction with B/F/TAF was attributed to seven main factors: *lack of side effects* (11/16), *improved health* (7/16), *being able to set the daily time for treatment taking* (7/16), *an easy treatment regimen* (6/16), *quick access to treatment* (6/16), *cost-covered treatment* (3/16), and *taking control of HIV* (3/16). However, most participants (11/16) discussed “having concerns with starting treatment”. Specifically, participants indicated *fearing side effects in the short- and long-term* (8/16) and that *taking treatment for life was daunting* (5/16). Half of the participants expressed “needing reassurance about treatment safety” (8/16) before feeling comfortable enough to begin taking it, and/or after experiencing initial side effects. Interestingly, seven participants noted “dissipating side effects over time” after one week of treatment. 

#### 3.2.4. Retention in Care

Eight categories were ascribed to the retention in care step of the HIV care cascade. All participants (16/16), at some point during their journey between being linked to HIV care and achieving viral suppression, discussed “facing psychosocial or health-related challenges beyond HIV”. These challenges included: *difficulty obtaining legal status in Canada and navigating the immigration process* (12/16), *difficulty accessing healthcare for issues other than HIV* (12/16), *a lack of income* (12/16), *social isolation* (13/16), *fearing COVID-19 infection* (11/16), and *difficulty integrating into Canadian society* (8/16). 

Almost all participants (15/16) discussed “feeling empowered to self-manage HIV”. They discussed three factors that contributed to this feeling: *receiving education about managing HIV from healthcare professionals* (13/16) (e.g., healthcare professionals answering HIV-related questions, explaining biomedical test results, and providing health advice), *receiving reassurance about living with HIV* (13/16) (e.g., participants feeling consoled by healthcare professionals about living with HIV), and *receiving relevant resources to manage health* (6/16) (e.g., healthcare professionals providing information about social and HIV-specific organizations that participants can access, as well as educational websites for reliable HIV information). Relatedly, most participants (12/16) also expressed their experience with “humanizing clinical encounters”. These encounters were described along six dimensions, five of which remain the same as those described at linkage: *feeling supported and cared for* (9/16), *feeling kindness from healthcare professionals* (10/16), *feeling safe and comfortable* (10/16), *building a patient-provider relationship based on trust* (7/16), *feeling heard and accepted* (8/16), and *feeling respected* (7/16). 

Most participants (11/16) noted that they were “enjoying smooth operations in the clinic”, which consisted of five factors: *accessible healthcare professionals* (9/16), *easy access to free care* (8/16), *care coordination issues often dissipated and/or addressed* (8/16) (e.g., patients’ confusion around navigation within the clinic diminished as they became more familiar with the system or after receiving clarification from healthcare professionals), *easy appointment bookings* (7/16), and *ability to speak with healthcare professionals in one’s native language* (4/16) (which, in these cases, were English, French, and Spanish). 

Most participants (12/16) described an experience of “receiving holistic care” through their multidisciplinary team. This was discussed by the participants as care that addressed their bio-psycho-social needs and sometimes expanded beyond their HIV-related health concerns (e.g., immigration and mental health related support). In this regard, participants expressed the importance of complementary care provided by the different clinicians on their team. Notably, patients mentioned turning to their doctors for their biomedical healthcare needs (e.g., questions around HIV management) and to social workers for their psycho-social needs (e.g., questions around food security or financial challenges). In fact, nine participants specifically highlighted the importance of the social workers in assisting with their immigration process, dealing with financial challenges, or being linked to other community organizations or services.

Two-thirds of the participants (11/16) discussed “wanting more frequent contact with healthcare professionals” through four means, in particular: via *telecommunication* (10/16), *outside of regular work hours* (4/16), via *more appointments* in general (4/16), and via more appointments specifically *with the social worker* (4/16). Most participants (11/16) mentioned “dealing with HIV-related psychological distress” (e.g., depression or fear of immigration rejection due to their HIV). Over half of the participants (9/16) also indicated “sharing responsibility to manage HIV” with healthcare professionals. All nine participants discussed specific *healthcare team duties*, which included: *providing clear explanations and guidance* (8/16), *creating kind and safe environments* (7/16), *ensuring that patient health improves* (6/16), *helping with navigation* (5/15), and *providing medication* (4/16). Eight patients discussed *patient duties,* which included: *self-managing HIV care and treatment* (8/16), *following clinicians’ instructions* (6/16), *asking questions* (5/16), and *attending appointments* (3/16).

#### 3.2.5. Adherence to Treatment

Two categories were ascribed to the treatment adherence step of the HIV care cascade. Two-thirds of the participants (11/16) expressed “being satisfied with treatment” beyond the first few days of starting treatment. Their satisfaction with B/F/TAF was attributed to seven factors: *improved health* (8/16), *an easy treatment regimen* (7/16), *taking control of HIV* (6/16), *quick access to treatment* (6/16), *a consistent supply of treatment* (5/16), *a lack of side effects* (4/16), and *cost-covered treatment* (4/16). Two-thirds of the participants (11/16) also mentioned “feeling resilient and responsible”, which was discussed as important in facilitating their sustained adherence to treatment. This feeling was most often identified as coming from *a desire to control HIV* (6/16) and *to protect others* (4/16).

#### 3.2.6. Viral Suppression

One category was ascribed to the viral suppression step of the HIV care cascade. Five participants expressed “finding more peace of mind since becoming undetectable”. Participants highlighted that, alongside feeling relieved, they also felt that this milestone confirmed their discipline and control over HIV.

#### 3.2.7. Perceived Health-Related Quality of Life (HrQoL)

Five categories were ascribed to the perceived HrQoL step of the HIV care cascade. Most participants (12/16) discussed “being helped by a supportive social network” beyond their healthcare team. Most (11/16) mentioned “wanting a long, healthy, and normal life”. Relatedly, over half (9/16) indicated “deciding to improve lifestyle habits since diagnosis”. Strategies to improve their lifestyle habits included: *being more mindful and/or careful with their health and wellbeing* (7/16), *eating healthier* (7/16), *exercising more* (5/16), and *taking more time for self-reflection and/or self-care* (5/16). Furthermore, half (8/16) mentioned “feeling better physically and mentally since starting care and treatment”. Finally, seven participants expressed that they were “fostering quality-of-life through activities” such as focusing on their occupation, education, and/or hobbies.

## 4. Discussion

This study explores the experiences of 16 MLWH enrolled in a prospective cohort study in Montreal, Canada, where B/F/TAF was being initiated free-of-charge and as soon as possible after linkage to multidisciplinary HIV care. To our knowledge, this is the first study that provides qualitative insights on the experiences of MLWH enrolled in such a model of primary HIV care. Our framework analysis yielded 30 categories of shared experiences by MLWH throughout their journey across steps of the HIV care cascade.

### 4.1. Diagnosis: Dominated by Distress & Immigration-Related Concerns

When discussing diagnosis, the most common experience that MLWH described was “initially experiencing distress”, followed by “questioning the impact of HIV diagnosis on immigration”. Migrants, in general, often experience mental health issues (e.g., depression, anxiety, post-traumatic stress disorder) and immigration-related challenges (e.g., language-barriers, difficulty with integration in a new country, issues with acquiring immigration status and health coverage) when moving away from their home countries [47,48,49,50,51]. In the context of Montreal, Canada, migrants are reported to experience substantially greater unmet healthcare needs compared to Canadian citizens with sufficient health coverage [52]. The intersectional burden of living with HIV and as a migrant has also been discussed as an element that potentially amplifies challenges, such as obtaining support for mental health and social care [53,54,55], the experience of racialized discrimination and stigmatization [53,54,55,56], and adversities during resettlement [54,55,56]. Also, the HIV and migrant co-status can lead to health coverage challenges (e.g., for international students), as not all insurance providers cover HIV care and treatment and getting access to public health insurance can be a major challenge, or not possible, when migrants are waiting for or transitioning between immigration statuses in Canada. These findings stress the value of embedding mental health and immigration-related support in primary HIV care settings, as well as providing cost-covered treatment for all.

### 4.2. Linkage: A Time of Navigation Challenges and an Opportunity to Connect with Clinicians

When discussing linkage to care, almost all MLWH spoke about “navigating the healthcare system with difficulty”. This issue is a well-documented challenge for migrants in general [57,58,59]. Moreover, access to family doctors and primary care has been a historic challenge in the province of Québec (where Montreal resides) [60]. For many MLWH in Montreal, the Immigration Medical Exam is often the first experience these people have with the healthcare system. Embedding a patient navigator (or staff with similar responsibility) may be a viable solution to helping patients efficiently transition between where diagnosis occurs and their HIV care services [61,62].

At this step, MLWH also expressed the importance of experiencing “humanizing clinical encounters”, where they perceived care, kindness, acceptance, respect, and safety from their healthcare professionals. Most of the MLWH also noted “being reassured about living with HIV”, “receiving personalized health information”, and “quickly accessing care”. Such experiences, the MLWH explained, were necessary as they fostered a sense of relief, alleviated major fears (e.g., of death), heightened their willingness to initiate treatment, and partially motivated their sustained engagement in care and treatment. These findings support the importance of providing care with respect and empathy. Previous studies highlight that, when clinicians adopt such approaches, they can better promote rapport-building, higher quality of care, and higher levels of medication self-efficacy [63,64,65]. 

### 4.3. Treatment Initiation: Rapid ART Is Satisfying, but Concerns Exist

When discussing treatment initiation, almost all MLWH expressed “being satisfied with treatment”, particularly due to a lack of side effects, improved health, an easy treatment regimen, and quick access to treatment. These findings support the provision of ART as soon as possible—an approach to care which is now possible due to major advancements in ART safety, ease-of-use, tolerance, and genetic barriers to resistance [35,36]. A potential barrier to rapid ART initiation may be the belief that patient-preparedness is critical to achieving ART readiness, however, scholars discuss how this long-held belief may be potentially harmful and non-evidence-based [66]. Furthermore, humanizing clinical encounters across healthcare settings (e.g., at the site of diagnosis) and efficient linkage to care (e.g., via care coordinators) may assist with patient-preparedness—though this must be further studied.

While rapid ART initiation has been demonstrated in the literature as feasible and well received, our findings suggest that rapid initiation must be done with caution, as most MLWH discussed “having concerns with starting treatment”, particularly around side effects. Half of the MLWH noted “needing reassurance about treatment safety” from their clinicians to feel comfortable with treatment, especially if they experienced side effects. Providing reassurance about treatment safety at initiation may further contribute towards building trust with healthcare professionals, which is necessary for addressing feelings of anxiety and vulnerability that may be experienced by MLWH when first enrolling in care and treatment [67].

### 4.4. Retention in Care: The Burden of Challenges beyond HIV and Importance of Patient-Centered Care

While being retained in care, all MLWH described “facing psycho-social or health-related challenges beyond HIV”. As found in other studies, people living with HIV encounter critical issues beyond their infection (such as lack of income, obtaining legal status, social isolation) that must be addressed [22,68,69]. However, most MLWH described “receiving holistic care”, where the multidisciplinary team of clinicians was able to assist MLWH in addressing their bio-psycho-social concerns, both in relation to HIV and beyond. Social workers were particularly praised for their assistance with psycho-social challenges and some MLWH expressed “wanting more frequent contact” with these healthcare professionals. This finding supports previous work that highlights the importance of multidisciplinary models for primary HIV care with sufficient funding for care providers such as social workers [22,70,71,72,73,74,75,76]. Although the act of embedding social workers into primary care services is becoming more common in Canada [76], specific training in HIV and immigration support may be necessary to ensure their comfort and capacity to work with this population [77,78].

At this step, MLWH also expressed that they were “enjoying smooth operations in the clinic”. It seemed that the longer they were engaged, the more comfortable they felt with navigating the service. Essential though for most MLWH was experiencing “humanizing clinical encounters”. They expressed how, beyond initial linkage to HIV care, warm encounters were important in sustaining motivation to remain engaged in care and were necessary for building a relationship based on trust with healthcare professionals, including non-clinician staff. Additionally, most MLWH explained that they were “sharing responsibility to manage HIV” with their healthcare professionals. Relatedly, almost all MLWH emphasized the importance of “feeling empowered to self-manage their HIV”, an experience which the MLWH noted as being fostered by increased comfort with and understanding of their treatment, its safety, and living with HIV, in addition to the resources and support systems available to them. These findings corroborate previous research that recommend “patient-centered” or “person-centered” approaches to HIV care [79,80,81,82,83,84]. Such approaches to care aim to ensure that patients have a functional and meaningful life and thus strive to incorporate empathy, respect, engagement, shared-decision-making, safety, trust, a holistic focus, and coordinated care as central tenets [84,85].

### 4.5. Adherence to Treatment: The Heightened Importance of Taking Control over HIV

Beyond treatment initiation, MLWH continued to express “being satisfied with treatment”. However, satisfaction at this step was attributed more to taking control of HIV and was much less focused on a lack of side effects. Additionally, most MLWH mentioned “feeling resilient and responsible”, which was discussed as important in facilitating their sustained adherence to treatment. This feeling was most often identified as coming from a desire to control HIV and to protect others. These findings suggest the value of promoting HIV self-management strategies among MLWH. Previous studies highlight the importance of HIV self-management for maintaining and/or improving ART adherence, HrQoL, and self-efficacy [86,87,88]. However, for self-management to occur, patients must be empowered, potentially via skills training and counselling [88]. Thus, promoting self-management approaches to care may be a secondary outcome of adopting patient- and person-centered care strategies.

### 4.6. Viral Suppression: Characterized by Peace of Mind and a Sense of Control

The clinical and public health basis for the importance of HIV viral suppression has been strongly emphasized in the literature [89]. However, the experiences of people living with HIV who have achieved viral suppression have been less explored. The results of this study indicate that MLWH who achieved viral suppression expressed “finding more peace of mind since becoming undetectable”. Indeed, by reaching this step, MLWH felt a sense of relief and control over their HIV infection. These feelings may encourage continued engagement with care and treatment. However, reaching this step may also present a shift in patients’ priorities (e.g., from HIV control to addressing other life stressors). Further research understanding MLWH needs at this step, alongside barriers and facilitators experienced here, is necessary.

### 4.7. Perceived Health-Related Quality of Life: Promoted by Social Networks and Personal Activities

Little research has been published on the lived experiences of MLWH around HrQoL [90]. Calls for patient- and person-centered approaches in HIV are advancing care and research priorities beyond viral suppression and undetectability to include optimal HrQoL [91]. This element could thus be evaluated, both quantitatively and qualitatively, throughout each step of the HIV care cascade [91]. In this study, qualitative findings suggest that the largest contributor to HrQoL for MLWH was “being helped by a supportive social network”. Informal social support networks have been well reported in the literature as important in improving the psychological wellbeing of people living with HIV [22,92,93]. Studies suggest that, after being diagnosed with HIV, partners, family members, and friends can motivate individuals to get linked to care and treatment, as well as assist in the health system navigation process [22,92]. These members can also provide emotional relief to MLWH, which in turn can positively impact their treatment adherence [22,92]. Most MLWH in this study also discussed “deciding to improve lifestyle habits since diagnosis”. It seemed that, the more the MLWH gained confidence and saw improvement in their health from engagement in care and treatment, the more careful many of them tried to be with their health and wellbeing. Lastly, MLWH highlighted the importance of “fostering quality of life through activities”, such as work, higher education, and hobbies. However, several factors (e.g., lack of a work visa, lack of knowledge of opportunities and resources available to them, and lack of proficiency in their host country’s language) can make engaging in such activities challenging for MLWH [22]. This further suggests the importance of embedding social workers or similar professionals in HIV primary care settings to support patients in these areas. 

### 4.8. Strengths and Limitations

A major strength of this study is the longitudinal nature of the data collection process. By conducting interviews at 1, 24, and 48 weeks after treatment initiation, a richer exploration of the experiences of MLWH over time was enabled. Further enriching the data was the sample’s diversity, notably in terms of age group, birth country, sexual orientation, and health coverage, as well as employment and immigration statuses. Moreover, a stakeholder engagement approach was taken, whereby the developed qualitative data matrix was validated with two patient partners, one research coordinator, and one research nurse. However, this study was conducted in one site (i.e., a quaternary hospital-based HIV clinic) in a high-income country, which may hinder generalization of findings. Another limitation is the small number of female interviewees, as few women agreed to join the cohort. This is a frequently encountered challenge in HIV clinical research, potentially due to patient distrust of researchers, competing family responsibilities, linguistic barriers, HIV-related stigma and perceived discrimination, and transportation difficulties [94]. Since this study requires people to enter a research process right after diagnosis, a potential bias in this study may be that the population who agreed to participate is more prone to engage in care and more ready to adhere to medication. It is also necessary to note that healthcare utilization patterns and challenges encountered by migrant populations can vary based on the amount of time they have spent in their new country [95]. Although many participants in this study were newly arrived migrants (arriving in Canada approximately two months prior to enrolling in this study), their duration of stay in Canada was not explicitly assessed and was thus not considered in the analysis. A final limitation is that patient recruitment was severely hindered by the COVID-19 pandemic. Although data from 16 MLWH provided a solid exploration, more participants could have increased the depth of the findings (e.g., some specific experiences may not be represented). Nevertheless, the saturation of the main analytical categories, by number of contributing patients and interviews, was high (see Table 2). 

## 5. Conclusions

In conclusion, to our knowledge this is the first study to report a qualitative analysis in a longitudinal cohort study on the experiences of MLWH enrolled in multidisciplinary HIV care, where treatment was being provided free-of-charge and initiated as soon as possible after linkage to care. In earlier stages of their HIV care trajectory, MLWH experience more negative emotions as a result of HIV-related distress, psycho-social challenges beyond HIV, and health system navigation challenges. However, efficient, humanizing, and holistic approaches to care, coupled with rapid ART initiation, seemed to help alleviate patients’ concerns, address their bio-psycho-social challenges, encourage their initial and sustained engagement with care and treatment, and ultimately contribute to positive experiences. While this study provides qualitative evidence for the value of multidisciplinary HIV care with cost-covered ART and its rapid initiation for MLWH, our findings suggest that this model must be sufficiently resourced and accompanied with patient- and person-centered care approaches.

## Figures and Tables

**Figure 1 jpm-12-01497-f001:**
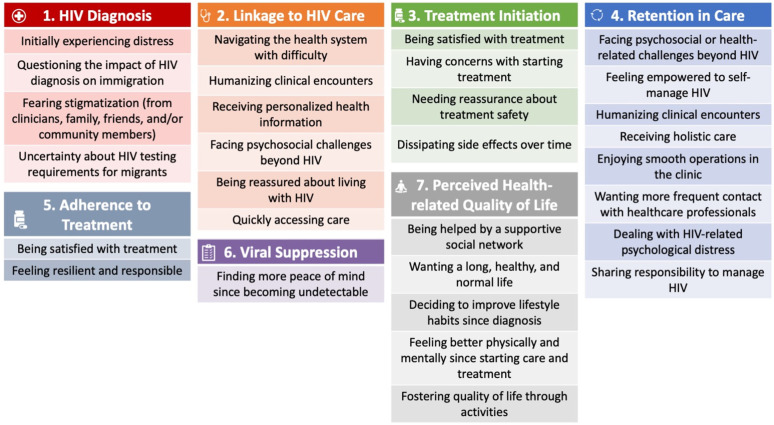
Main categories of MLWH experiences by step of the HIV care cascade.

**Table 1 jpm-12-01497-t001:** Sociodemographics at study enrollment of interviewed MLWH.

Participant #	Age	Sex	SexualOrientation	Region of Birth	Immigration Status	Health Coverage	Education	Paid Employment Status	Interviews Completed
Week 1	Week 24	Week 48
1	41–50	Male	Heterosexual	East Asia	Visitor	No or Low Coverage	College/CEGEP/Technical Degree	Unemployed			
2	21–30	Male	Bisexual	East Africa	International Student	No or Low Coverage	University	Unemployed			
3	21–30	Female	Heterosexual	Southern Africa	Asylum Seeker	Sufficient	University	Paid Employment			
4	51–60	Female	Heterosexual	Southern Africa	Asylum Seeker	Sufficient	College/CEGEP/ Technical Degree	Unemployed			
5	41–50	Male	Homosexual	Southeast Asia	Asylum Seeker	Sufficient	University	Unemployed			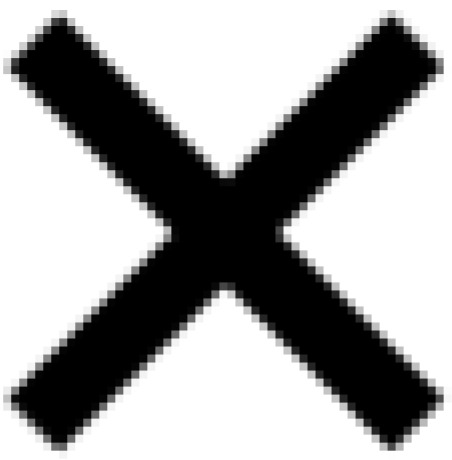
6	21–30	Male	Homosexual	North Africa	International Student	No or Low Coverage	University	Unemployed			
7	31–40	Male	Homosexual	Latin America	Asylum Seeker	Sufficient	College/CEGEP/Technical Degree	Paid Employment			
8	21–30	Male	Bisexual	North Africa	Temporary Worker	No or Low Coverage	University	Paid Employment		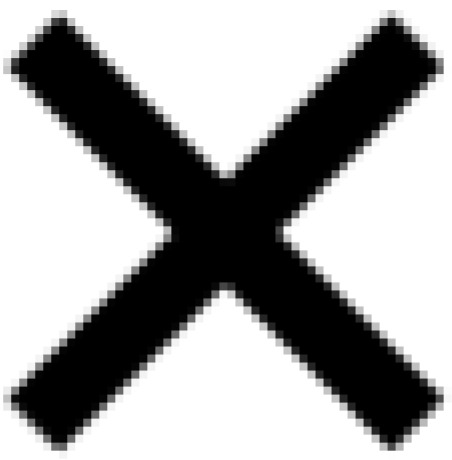	
9	31–40	Male	Bisexual	Southern Africa	Asylum Seeker	Sufficient	University	Paid Employment			
10	21–30	Male	Homosexual	Latin America	No Status	No or Low Coverage	University	Paid Employment		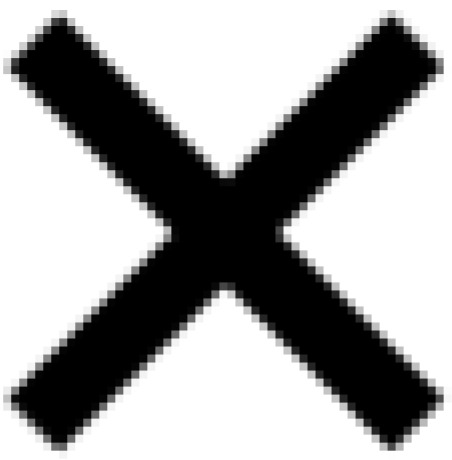	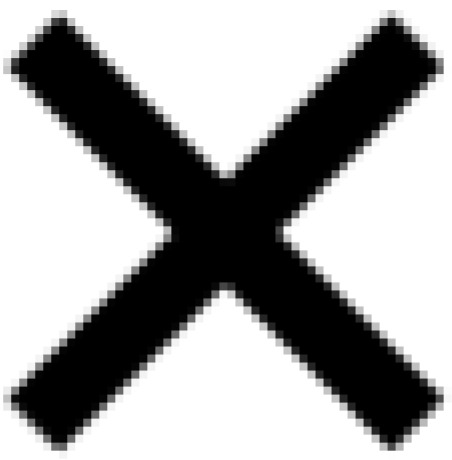
11	41–50	Male	Homosexual	Latin America	Visitor	Sufficient	University	Unemployed		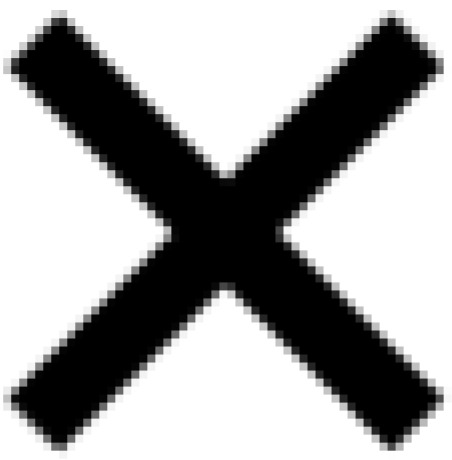	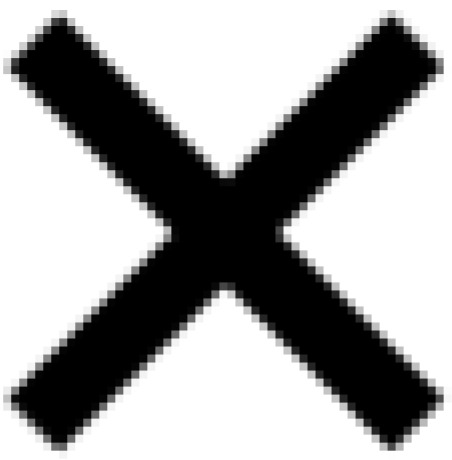
12	21–30	Male	Homosexual	East Asia	International Student	No or Low Coverage	Secondary/Professional Degree	Paid Employment		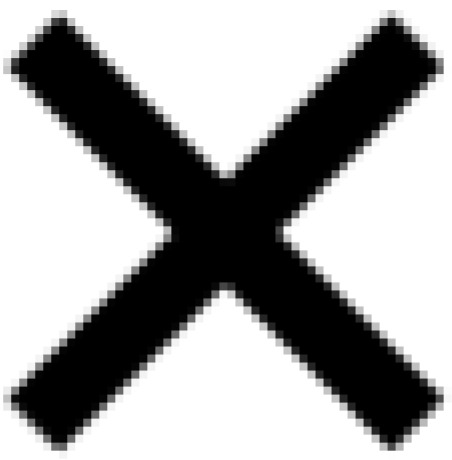	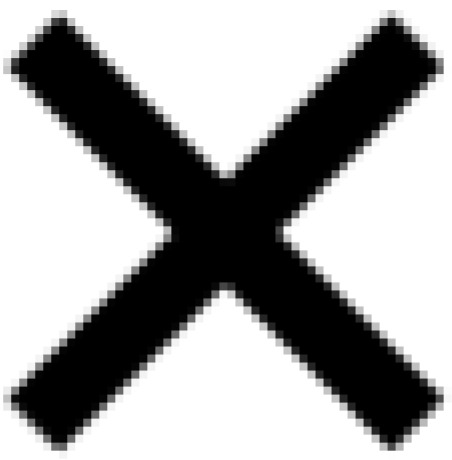
13	21–30	Male	Homosexual	Southeast Asia	Visitor	No or Low Coverage	Secondary/Professional Degree	Unemployed		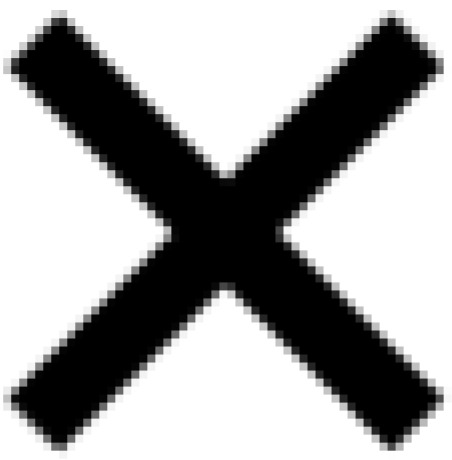	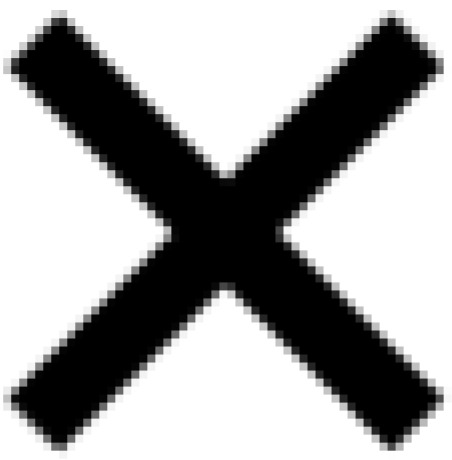
14	21–30	Male	Heterosexual	Caribbean	Asylum Seeker	Sufficient	College/CEGEP/Technical Degree	Unemployed		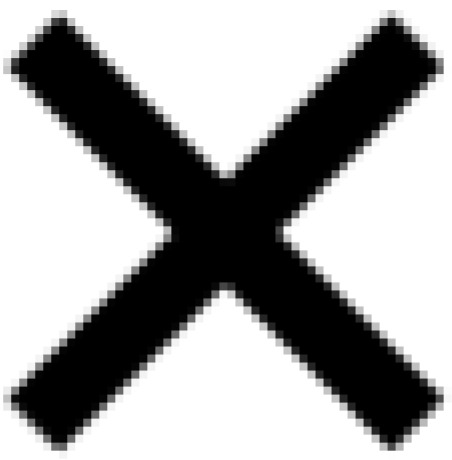	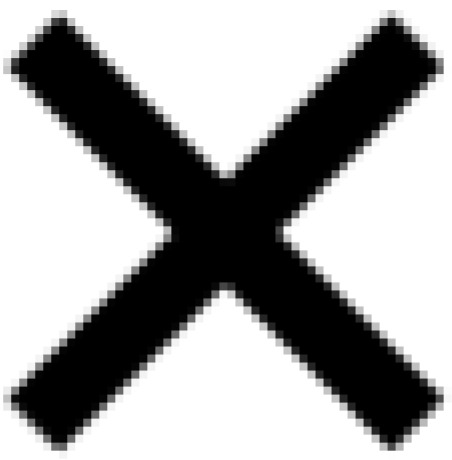
15	51–60	Male	Homosexual	Western Europe	Naturalized Citizen	Sufficient	University	Unemployed		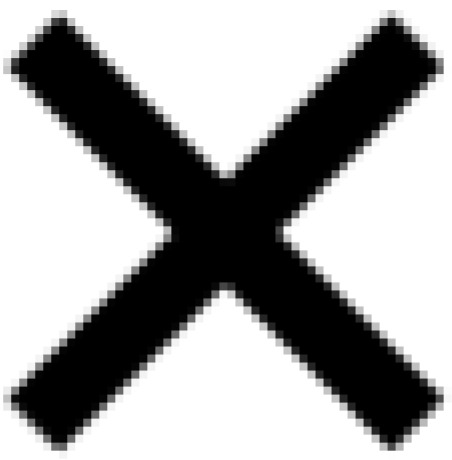	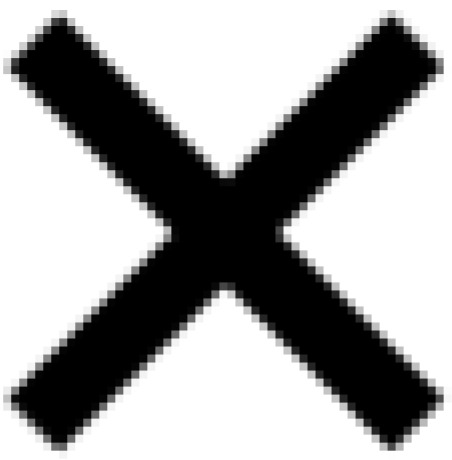
16	31–40	Male	Heterosexual	Caribbean	Asylum Seeker	Sufficient	University	Unemployed		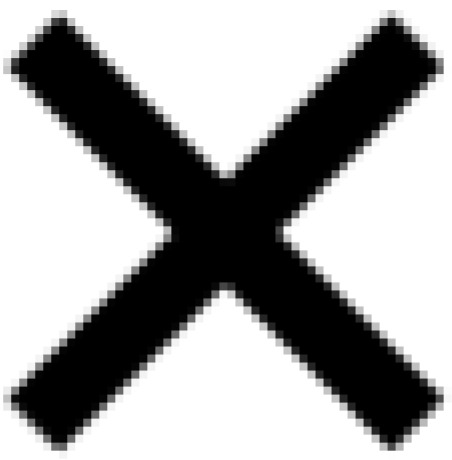	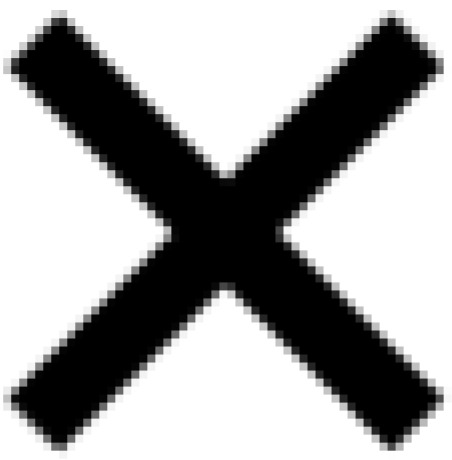

**Table 2 jpm-12-01497-t002:** Data framework matrix for MLWH experiences.

Category	Illustrative Excerpt	Contributing Participants (n/16)	Contributing Interviews (n/32)
HIV Diagnosis			
Initially experiencing distress Feeling: - Worried and/or scared - Shocked - Confused - A loss of control over their life	“I could not believe it. I came [to Canada] with hopes, I had a dream. And I did not believe it […] I blacked out.” Participant #5, W1	10 7 6 6 4	14 8 7 6 4
Questioning the impact of HIV diagnosis on immigration	“I was very concerned about how the diagnosis might impact my permanent residency process. I thought I had to go back to my country.” [Translated from French]—Participant #8, W1	6	6
Fearing stigmatization (from clinicians, family, friends, and/or community members)	“I just worried about [how] to tell people around me, for the first thing.”—Participant #30, W1	4	5
Uncertainty about HIV testing requirements for migrants	“Normally, the work permit does not require a medical visit. […] It was not mandatory to have a medical examination. Me, I wanted to do [the examination] for if I find a volunteer [position] in a hospital [so] I can work without any problem…” [Translated from French]—Participant #6, W1	3	3
**Linkage to HIV Care**			
Facing psychosocial challenges beyond HIV - Mental health - Immigration - Securing finances and/or health insurance - Learning Quebec’s official language (French)	“I feel dreaded, maybe I wasn’t eating well. I was worrying about so many things that, so many financial challenges there… I haven’t seen doctors for months, years maybe, because I don’t [have] insurance or anything and I couldn’t afford it.”—Participant #4, W48	12 11 8 6 4	23 16 10 10 4
Navigating the health system with difficulty - Across clinics and organizations - Within the CVIS	“You have to understand as a foreigner […] for all the [health] system running in North America, I have no clue, no idea … And then the language problem as well, ok … Because especially for me, I’m a foreigner. I don’t know the procedure [for accessing care] or the round, you know.”—Participant #1, W1	15 15 9	21 20 12
Humanizing clinical encounters Feeling: - Supported and cared for - Kindness from healthcare professionals - Safe and comfortable - Heard and accepted - Respected	“[The care I received] was perfect, I felt loved, cared for. I felt understood, for the first time. Everything that happened to me was not planned. I did not have a plan to take care of this. Then, when I arrived here, I had it, I had a plan. I met Dr. [name omitted] and the other people and they told me it was important to start care, and they told me how it would happen. I never felt any safer than I felt around these people. I felt helped […]”—Participant #4, W1	14 12 10 7 5 3	19 15 12 7 6 3
Being reassured about living with HIV	“Well, before I had misinformation, bad ideas about this disease. But when I came to the hospital, [the health professionals] calmed me down. They said to me, ‘There’s nothing to worry about. It’s just that you’re going to have treatment and then you’re going to be fine and you’re going to live your normal life.’ With their behavior, how they talk to me, all that was good. I left the clinic really happy. […] There was a big difference between how I entered the clinic and when I left the clinic.” [Translated from French]—Participant #6, W1	12	19
Receiving personalized health information	“And also, I’m always asking like: ‘My blood pressure, is that a good thing?’ And they are like: ‘Yeah, yeah, that’s good.’ And I always have questions and they answer very well … Because it’s not like, you know when you ask a question and then somebody gives you one answer one way. No, they actually explain.”—Participant #3, W1	13	17
Quickly accessing care	“In fact, it went very quickly, I received a call telling me that I had to show up here. I was given the news and it was very difficult to take at that time. But very quickly, I think the next day or two, I had an appointment. And I met everyone, the social worker, the nurse, the doctor.” [Translated from French]—Participant #2, W24	10	12
**Treatment Initiation**			
Being satisfied with treatment Due to: - Lack of side effects - Improved health - Being able to set the daily time for treatment taking - An easy treatment regimen - Quick access to treatment - Cost-covered treatment - Taking control of HIV	“The medicine is really good, it’s really great because I don’t feel bad at all. I feel fine, no pain, nothing.” [Translated from French]—Participant #14, W1	15 11 7 7 6 6 3 3	18 11 8 7 7 6 4 4
Having concerns with starting treatment - Fearing side effects in the short and long-term - Taking treatment for life was daunting	“[Starting treatment] was a hard decision because [when] you start, you cannot stop to take [the treatment]. But you cannot avoid the situation, you have to take it. So, you have no choice. So, personally it’s hard because I like to choose everything that I do but, in this case, I don’t have any option.”—Participant #7, W1	11 8 5	15 9 8
Needing reassurance about treatment safety	“The only thing for me is dizzy[ness]. Especially for the first day … So, I talked [about] this to Dr. [name omitted] again. He said: ‘The body needs time for the medication.’ So, it’s have to be take time.”—Participant #1, W1	8	13
Dissipating side effects over time	“Well the first 2 days I was in a lot of pain. I think, like, my body was getting used to it, but I was really nauseous. And I had nightmares. A lot. […] it was intense at first, but now it’s okay.” [Translated from French]—Participant #2, W1	7	7
**Retention in Care**			
Facing psychosocial or health-related challenges beyond HIV - Difficulty obtaining legal status in Canada and navigating the immigration process - Difficulty accessing healthcare for issues other than HIV - Lack of income - Social isolation - Fearing COVID-19 infection - Difficulty integrating into Canadian society	“Yeah, not having enough money, that is a barrier. So, I couldn’t even go, if I wanted to go get some fruit or something, the money would challenge me. I would just stay and do with whatever I had. […] Financially, if there is some organizations that can help give you something like a coupon to go get some food at [the grocery store], whatever, I would welcome that. But I don’t have that kind of access.”—Participant #4, W48	16 12 12 12 13 11 8	31 22 20 18 17 13 12
Feeling empowered to self-manage HIV - Receiving education about managing HIV from healthcare professionals - Receiving reassurance about living with HIV - Receiving relevant resources to manage health	“I got control of my health. So, if I come here then I understand: ‘Ok, my CD4 count is 715.’ Then I know: ‘Ok, now I know that [I’m] ok. I’m a step ahead. My health is excellent, so I have to maintain it.’ And what is CD4 count? That’s what I’m going on Google. On Google ‘what is CD4 count’. Then I know: ‘Ok. These are the white blood cells and bla, bla, bla. And what is viral load?’ That’s how I do it. So, yeah, I feel good about it. I feel like I’ve got control of my health.”—Participant #9, W48	15 13 13 6	31 29 21 6
Humanizing clinical encounters - Feeling supported and cared for - Feeling kindness from healthcare professionals - Feeling safe and comfortable - Building a relationship based on trust with healthcare professionals - Feeling heard and accepted - Feeling respected	“I think I like the attitude of the staff. You know, they’re always like happy and excited to see you and talk and listen. It’s more like they’re concern[ed] about, you know, for you as a person not just like as in a patient. Ok, you know, looking at the time. It’s not like that. It’s like they have time for you. I think that’s really good because, you know, you don’t feel like you’re inconveniencing people or anything like that. So, that makes me look forward to the visits and also all the questions I have, they get answered and they get explanations. Because naturally, I’m anxious on my health questions and things, and I always get them answered.”—Participant #3, W24	12 9 10 10 7 8 7	25 21 20 15 14 12 9
Enjoying smooth operations in the clinic - Accessible healthcare professionals - Easy access to free care - Care coordination issues often dissipated and/or addressed - Easy appointment bookings - Ability to speak with healthcare professionals in one’s native language	“Personally, I find that the system you have adopted, especially for follow-ups with foreigners without [provincial health insurance], is really effective. […] And I also like the fact that the main person I come into contact with is either [the study coordinator] or [their HIV physician] only because they are the main people that are directly related to care, and who I think are, for this team, the main players in what you call caregivers. I like the format that even if I know there are people who are ‘back-up’ like social workers or nurses, there are still only two people who come into contact with me. Because, from the moment there are too many people who intervene, it is more difficult to manage, you see. And I think that precisely for a patient, it is not what he would want that there are too many things to do. I think that’s just like enough for it to be effective…” [Translated from French]—Participant #2, W48	11 9 8 8 7 4	23 16 14 14 8 4
Receiving holistic care	“[The healthcare professionals] have different responsibilities because, you know, they all have different experience in their professions. So, like, for example, I have a social worker who can help me like: ‘Oh, you can go to this if you need food, there are food banks or this, this.’ And then, the doctor will tell you about like, you know, what questions I have about health and that’s good. So, it’s like they both have different… Everybody has their own [role]. Just like, you know, how the body like the head has its function and the hands has its function, I feel like it’s like that. And then together they make like a complete.”—Participant #3, W48	12	20
Wanting more frequent contact with healthcare professionals - Via telecommunication - With the social worker - Outside of regular work hours - Via more appointments	“I know that you guys are busy but maybe when I go after a month or so, just text, email: ‘How is everything?’, whatever. It would also add more to my confidence as well, knowing there are people out there.”—Participant #4, W48	11 10 4 4 4	20 16 6 6 5
Dealing with HIV-related psychological distress	“Taking medication is important to physical health, but my social and mental health is still not good.”—Participant #5, W24	11	17
Sharing responsibility to manage HIV - Healthcare team duties - Patient duties	“I feel 100% responsible. I’m on top of my game. I’m doing what’s right. I don’t forget. I don’t need an alarm, my brain I programmed it. It’s [a] mindset […] [The healthcare professionals are] 200% plus responsible for all this, yes. They’ve helped me a lot in achieving [undetectability], brought my confidence […] Everybody, the whole team involved in this, I appreciate what they have done. They have made me feel comfortable. They’ve never made me feel any different. Like I’m when I walk in, I’m like I’m coming home. So, this has really driven me to commit to it. If I [was] feeling judged or didn’t feel wanted, or looked at in a different way I wouldn’t have committed. So, they’ve helped a lot.”—Participant #4, W48	9 9 8	9 9 8
**Adherence to Treatment**			
Being satisfied with treatment Due to: - Improved health - An easy treatment regimen - Taking control of HIV - Quick access to treatment - Consistent supply of treatment - Lack of side effects - Cost-covered treatment	“Oh, yeah, yeah, yeah! Absolutely! Yeah, there are a lot of changes. I feel energetic. I don’t feel that fatigued. I feel confident. I see life with HIV. So, yeah definitely things, they have changed […] on the psychological side, it’s been so positive […] now I feel much better. My emotions they’re not as how they were before. So, yeah, I feel much better […] now I feel more calm. I feel like I’m at the right place. I’m getting the right treatment…”—Participant #9, W24	11 8 7 6 6 5 4 4	23 14 10 9 7 7 5 5
Feeling resilient and responsible - From a desire to control HIV - From a desire to protect others	“I quickly got into the habit of taking [my HIV medication] because that’s what I can do to keep my partner healthy and safe. So I take it and for me it’s positive, it allows me to keep discipline and control over what’s going on. […] I don’t think I forget, or else it happens very rarely […] Then too, there is my discipline. I’m studying and working, so I can’t, I don’t have time to think about it, I maintain my discipline and I take my medicine and go to consultations, and the team is there for me too. So I don’t think about the disease anymore, I do what I have to do and I don’t have to think about it. It’s just a routine for me.” [Translated from French]—Participant #2, W24	11 6 4	18 8 4
**Viral Suppression**			
Finding more peace of mind since becoming undetectable	“Now it’s more quiet like more relaxed […] It’s like less anxiety […] Because now I know I’m undetectable so, it makes me feel like: ‘Ok, you are doing it well. It’s part of your routine. So, you are like well disciplined. So, you are doing something good for yourself.’ So, it’s a big difference. Like when I start, I was scared like: ‘Oh, maybe I’m not capable but I have to do it. I need to try it.’ And now I know I’m capable so it’s like: ‘Ok. It’s a really, really big change.’”—Participant #7, W24	5	6
**Perceived Health-related Quality of Life**			
Being helped by a supportive social network	“Everyone around me just like told me to live a stronger and don’t think so much. They always support me […] Because I have a few close friends that I [can] talk [with]. Yeah, so everyone like [comforts] me and yeah, excepts [me].”—Participant #13, W1	12	16
Deciding to improve lifestyle habits since diagnosis - Being more careful with their health and wellbeing - Eating healthier - Exercising more - Taking more time for self-reflection and/or self-care	“My quality of life is getting better because now I’m conscious. I was living carelessly and I cannot do it anymore. So it will improve my quality of life. This is a lifetime process and I need to make changes. I am reading books and information to know if I eat right so my immune system is helping me. I know this is all for the better.”—Participant #4, W24	9 7 7 5 5	15 10 8 7 6
Wanting a long, healthy, and normal life	“I hope to be healthy and be able to live normal. I still have my hopes and dreams and I hope that the treatment will help me achieve them. I just wanna be healthy and normal. I don’t ask for [more] benefits.”—Participant #5, W1	11	14
Feeling better physically and mentally since starting care and treatment	“In fact, everything that I had a problem with related to my physical health was related to the virus. […] Because I was diagnosed, I think, a little too late. […] so when I started taking the medicine, well all those little things that were bothering me [with] my physical health went away. So inevitably my mental health has improved […]” [Translated from French]—Participant #8, W48	8	15
Fostering quality-of-life through activities	“I think I have a good quality of life. I work. I pay more attention to what I miss. I exercise. I run three times a week. I think I improved on that.” [Translated from French]—Participant #2, W24	7	8

## Data Availability

Data can be accessed upon reasonable request by contacting the first author.

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
