# Peer review of "Experiences of Migrant People Living with HIV in a Multidisciplinary HIV Care Setting with Rapid B/F/TAF Initiation and Cost-Covered Treatment: The ‘ASAP’ Study"

_jpm, 2022, doi:10.3390/jpm12091497_

Round 1
Reviewer 1 Report
It is an original study of interest both in the care of migrants and in the care of seropositive people. It is unusual to find qualitative studies, which are the most appropriate to assess the user experience.
I have some comments to improve the manuscript:
Plese provide the place where the study was conducted in the title or in the abstract.
Line 107: The people enrolled (n=49) were all migrant people, please clarify.
Results section:
The first paragraph it is fine, nevertheless, all data are presented in Table 1. Maybe there is no need to repeat the numbers in teh text and only describe with words.
The table 2 corresponds to what is expected from the presentation of resultas of a qualitatuive study. I am not an expert in qualitative research, but it seems to me that it is not appropriate to present the frequencies in the text in which the results are described. They have already prostrated them in table 2 and they take away reading fluency. It is not a quantitative study, therefore, the frequencies have no value.
It would be highly recommended to present the categories found in a diagram, in which the boxes are the stages of the care process and contain the categories for each stage. It is a way of visually presenting the results that helps to understand them.
Dioscussion and conclusions:
The results found and the conclusions are expected given the context of the study.
The researchers relate in a good way the limitations and strengths of the study. Certainly, being a longitudinal study, it allows us to appreciate the sequence of the care experience of migrant patients. They also make recommendations.
Reviewer 2 Report
Dear authors,
This is a very interesting topic, which you rigorously investigated. Yet, there are certain concerns that have to addressed:
1. The issue of translation and how you managed the associated problems. You mentioned that all interviews were conducted in English, French, and Spanish. What was done for quality assurance?
2. The duration of stay in Canada. The problems that immigrants face depend on how long they stay in the country. Did you ask this question? If yes, consider this in your analysis. If not, mention as a limitation.
